# Design and Analysis of an Inductive Coupling System for the Early Detection of Heart Failure

Raghavendiran Krishnamurthy Venkataramani [1], Krithikaa Mohanarangam [2], Jongmin Lim [3], Ke Yu [1], Venkateswarlu Gonuguntla [4] and Jun Rim Choi [1,3,*]

1    School of Electronic and Electrical Engineering, Kyungpook National University,
     Daegu 41566, Republic of Korea
2    Symbiosis Institute of Technology, Pune Campus, Symbiosis International (Deemed University),
     Pune 412115, India
3    School of Electronics Engineering, College of IT Engineering, Kyungpook National University,
     Daegu 41566, Republic of Korea
4    Symbiosis Centre for Medical Image Analysis, Symbiosis International (Deemed University),
     Pune 412115, India
*    Correspondence: jrchoi@ee.knu.ac.kr; Tel.: +82-053-950-6567

**Abstract:** Heart failure is a common, complex clinical syndrome with high morbidity and mortality. Hemodynamic parameter evaluation is useful for early detection, clinical outcome monitoring, timely treatment, and the overall prognosis of heart failure patients. Therefore, continuous monitoring of hemodynamic parameters helps in the evaluation of patients with suspected heart failure. The hemodynamic parameters change with respect to the contraction and expansion of the heart. Hence, in this research, two circuit-less 30 mm spherical receiver coils were implanted in both the left and right sides of the heart and an external transceiver coil was placed above the chest. The changes in the reflection coefficient of the transceiver coil at the resonant frequency changed depending on the distance between the implanted coils, allowing the contraction and expansion of the heart to be determined. This work was carried out at 13.56 MHz, considering the safety limits imposed by the FCC. The proposed reflection coefficient monitoring technique may distinguish healthy patients from heart failure and heart attack patients. The reflection coefficients at a maximum distance of 50 mm for simulation and measurement are −10.3 dB and −10.6 dB, respectively, at the resonant frequency.

**Keywords:** heart failure; hemodynamic parameters; resonant frequency; reflection coefficient; specific absorption rate

## 1. Introduction

Cardiovascular diseases are the leading cause of death in the world, and the number of people with heart failure is rising dramatically. Over the years, implantable medical devices have seen tremendous progress as they help with the early diagnosis of symptoms related to cardiovascular diseases. Therefore, there is an urgent need to develop an inductive coupling system that facilitates early diagnoses even before hospitalization. A significant amount of research is devoted to monitoring the hemodynamic parameters of the heart for the early diagnosis of heart failure. Some of the existing works are discussed in the following literature review.

### 1.1. Literature Review

In [1], an implantable coil was positioned in the left ventricular area at a depth of 3.5 cm for monitoring cardiac pressure continuously. This work focused primarily on inductive power transfer (IPT), but it also provided continuous monitoring of cardiovascular pressure and involved electronic circuitry within the human body due to its primary emphasis on IPT. A pressure sensor was incorporated into this design of an implantable medical

device to allow for constant monitoring of a broad blood pressure range of 5–300 mm of mercury [2]. This system is quite complex and is only suitable for measuring pressure within the heart. In [3], an implantable, wireless vascular electronic system, consisting of an inductive stent and printed soft sensors was designed for real-time monitoring of arterial pressure, pulse rate, and flow. Although this system works without circuits inside the human body, it is only capable of sensing blood vessels and is unable to monitor the ventricular chamber size. Wireless technologies were applied to wearable chemical sensors in the design of an antenna's RLC circuit in which an antenna's resonance frequency varies with analyte concentration. A wireless, compact, percutaneous near-infrared spectroscopic device for continuous monitoring of local-tissue oxygenation was created in [4], and the sole purpose of this work was to monitor the saturation level of oxygen in organ grafts and flaps. A self-tuned open interior micro-coil antenna was designed for blood vessel imaging, data telemetry, and efficient wireless power transmission in [5]. This work was geared towards implantable wireless magnetic resonance imaging (MRI), and it operates at a very high frequency of up to 920 MHz, which is quite harmful to patients as it focuses on the brain. In [6], a left ventricular assist device (LVAD) was autonomously operated for evaluation of the ventricular chamber size using sensors that have a relation between the transmission coefficient and spatial separation in a resonantly coupled regimen. Despite this work having a greater impact on determining the chamber size, it did not provide a method for determining heart rate and other hemodynamic parameters. These days, physiological functions including heartbeat, local stimulation, data recording, and even medication distribution can now be influenced by implantable medical devices. This includes high blood pressure telemetry systems, cardiac recorders, and defibrillators [7–9]. Although all of these works are deemed valuable, they involve complex procedures and active electrical components inside the human body that are uncomfortable and possibly harmful to the patient [10–12].

*1.2. Overview of the Proposed Method*

In this body of work, we have alluded to the concept of measuring heart size in order to determine a heart's optimal operating conditions. This work did not involve placing any active electrical components into the human body at any point in our process. The current state of the heart was monitored by inserting two flexible spherical receiver coils on either side of the ventricular chambers and positioning a transceiver coil directly above the chest, which was required to observe the changes in the reflection coefficient in the external coil at the resonant frequency. In comparison to existing works, this work not only provides a viable solution for measuring the size of the ventricular chambers by providing the distance between the coils implanted on both sides of the heart, but it also solves the problem of involving electronic circuitry in the human body.

The process that was used for the experimental part will also be used for the real-time application of our method. Whereas, the coils will be made of flexible PCB, rather than the copper coils that were used for the experiments, because the flexible PCB is very light and also provides better comfort to the patient. In addition, a nanoVNA will be attached to the transceiver or the external coil during the practical application of our system so that the system as a whole is quite compact and it will be convenient to measure the changes in the receiver coils.

The remaining sections in the paper are arranged in the following categories: In Section 2, the methodology and optimization procedure of the proposed inductive coupling system are discussed. In Section 3, the simulated results obtained using a heterogeneous phantom with varying distances between the implantable coils are described. This section also discusses the effects of frequency on animal and human tissue. The results of the fabrication and measurements are reported in Section 4 and, finally, the work is concluded in Section 5.

## 2. Methodology and the Optimization Procedure

The changes in the resonance frequency of the coils can be utilized to determine the distance between the coils and the size of the ventricular chamber. The size of the coils is determined by the dimensions of the heart. Ansys HFSS was used to optimize the coils with respect to turns, outer diameter, wire thickness, wire spacing, and pitch at an operational frequency of 13.56 MHz using the iterative optimization approach outlined in [13]. The transceiver ($Tx$) coil and receiver coil ($Rx_1$ and $Rx_2$) locations are depicted in the schematic diagram in Section 1, which may be found in Figure 1. It is abundantly clear from the figure that the receiver coils are positioned on the surface of the ventricular chambers of the heart. These coils transmit information to the transceiver coil that is positioned above the chest whenever there is a change in the distance between them. A VNA is connected to both ends of the $Tx$ coil so that the changes in the size of the heart can easily be monitored.

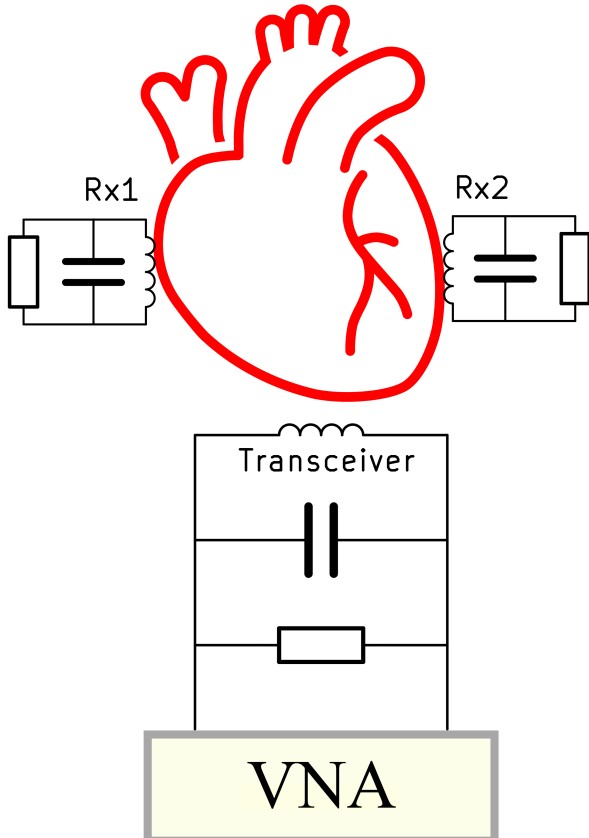

**Figure 1.** A schematic diagram representing the placement of the coils in and above the heart.

### 2.1. Equivalent Circuit Model of the Inductive Coupling Method

Figure 2 is the equivalent circuit model of inductive coupling, where, $V_{AC}$, $R_s$, $R_T$, $C_T$, and $L_T$ represent supply voltage, source resistance, stray resistance, stray capacitance, and primary inductance, respectively. The receiver side has similar components, such as secondary inductances ($L_{R1}$ and $L_{R2}$), stray capacitances ($C_{R1}$ and $C_{R2}$), stray resistances ($R_{R1}$ and $R_{R2}$), and load resistances ($R_{L1}$ and $R_{L2}$). Electrical resonance is a phenomenon that happens in an electric circuit at a specific frequency called the resonant frequency when the fictitious parts of the impedances or admittances of the circuit elements cancel each other out. A circuit's response is optimum at this specific resonant frequency. Any circuit with an inductor and a capacitor can experience resonance since both of these components have the capacity to store energy [14].

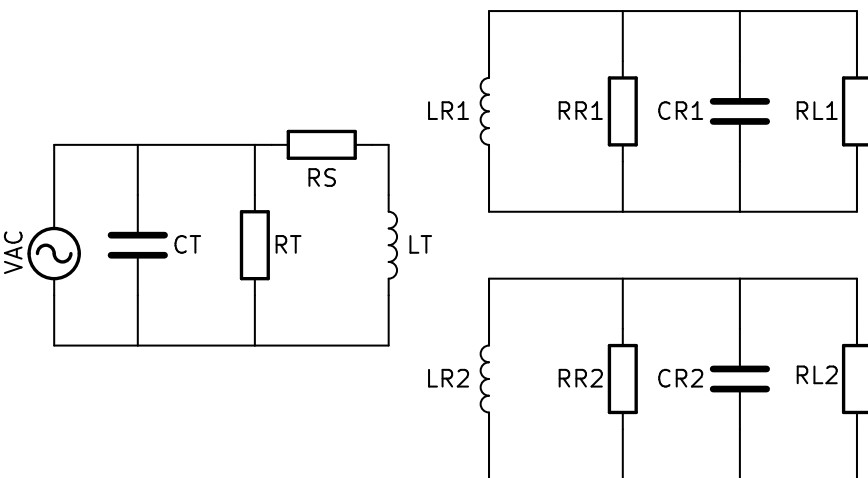

**Figure 2.** The equivalent circuit diagram of the proposed method.

Energy oscillates between the inductor and capacitor in a resonant circuit, and the rate at which energy is transferred between these two components relies on the values of L and C. We will observe oscillations in the circuitry as a result of these oscillations in energy transmission. If this circuit is perfect and there are no resistive parts of any type, these oscillations will never stop. However, in practice, there will be some resistance present in every circuit. These oscillations will deteriorate as a result of the presence of this resistance. We need to provide this LC circuit with power from an external source at the same frequency to maintain these oscillations (addressed as resonant frequency). We can continue the oscillations in this manner. The two primary fundamental resonance circuits are parallel and series RLC circuits. A concept called the quality factor (Q-factor) can be used in resonance circuits to gauge how sharp the resonance is [15]. A dimensionless metric called the Q-factor is used to determine how effectively a system oscillates. A high Q-factor means that oscillations have more energy and travel farther, while a low Q-factor causes oscillations to weaken more quickly. The frequency needs to be near the real resonant frequency in order to attain a high Q-factor (smaller bandwidth). Hence, a high Q-factor has more energy but is more challenging to tune [16].

The Q-factor is given in simple terms by Equation (1):

$$Q = 2\pi \times \frac{\text{maximum stored energy}}{\text{dissipated energy in one period at resonance}}. \tag{1}$$

The resonant frequency can be determined using Equation (2):

$$f_s = \frac{1}{2\pi\sqrt{LC}}, \tag{2}$$

where *L*, *C*, and $f_s$ represent the inductance, capacitance, and resonant frequency, respectively [1].

### 2.2. Optimization Procedure of the Proposed Method

To achieve ideal coil geometries, a design approach as depicted in Figure 3 was implemented, and the Ansys Electronics Desktop 2018.0 was employed to simulate and optimize $Rx_1$, $Rx_2$, and $Tx$. These steps were followed in the design process:

Step 1: The design restrictions are determined in accordance with the application. There must be a determination regarding the size and shape of the coils. Taking into account the form and size of the average human heart, the spherical receiver coils are selected with a maximum diameter of 3.4 mm (we selected two spherical receiver coils). As $Tx$ lies on the chest, a planar circular shape is chosen to ensure comfort. The values $Rx_1$, $Rx_2$, and $Tx$ are wound with copper, and the operational frequency is 13.56 MHz.

Step 2: In this step, the parameters of $Rx_1$, $Rx_2$, and $Tx$ are initialized. The parameters are the inner diameter of the coil ($d_I$), the width of the wire ($w$), the number of turns ($n$), the distance between turns ($s$), and the pitch ($p$), respectively.

Step 3: The mutual inductance and $Q$-factor between the coils are determined by performing a parametric sweep with regard to the parameters mentioned in the preceding step.

Step 4: The procedure for optimization is performed until both $Q$ and $M$ have increased to their maximum values. In order to determine the transmission coefficients, the capacitance is continually adjusted to 13.56 MHz.

Step 5: At the fifth stage, a comparison is made between the power transmission efficiency of the coils when operating in air and when operating in a tissue medium.

Step 6: Finally, the specific absorption rate, often known as the SAR, is calculated to ensure that the design complies with the requirements set forth by the Federal Communications Commission (FCC).

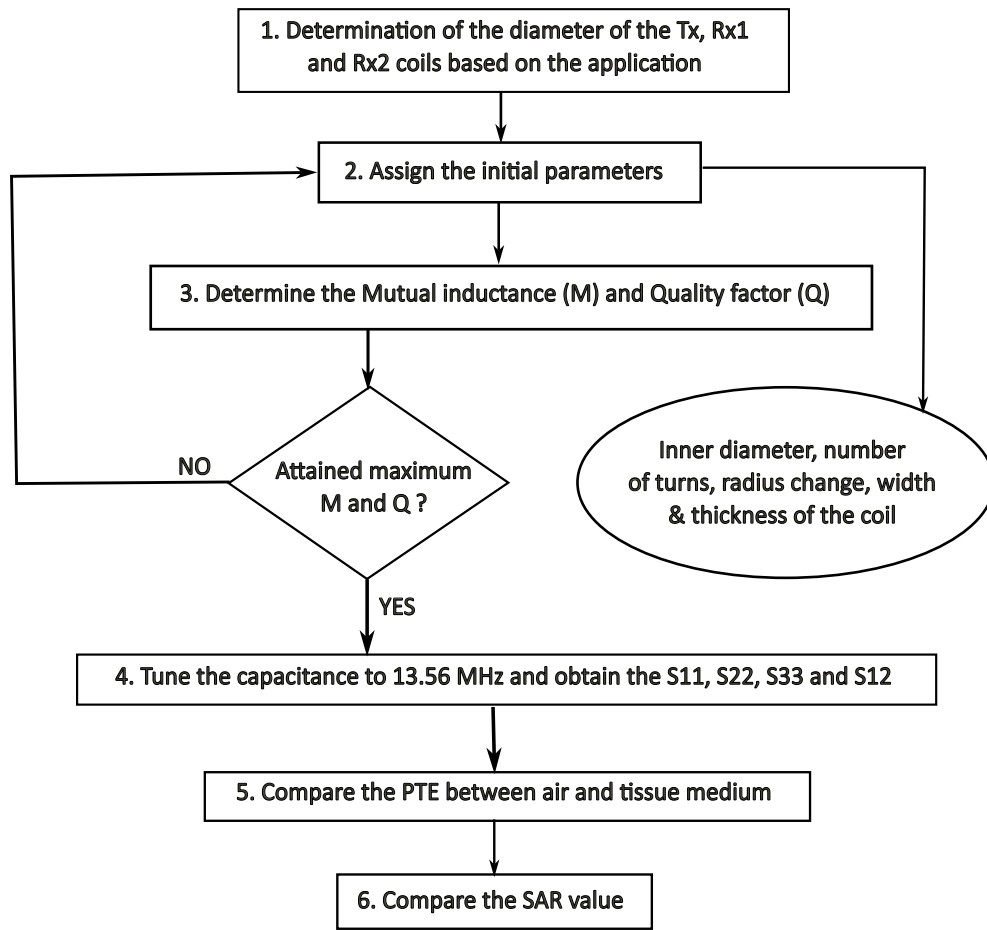

**Figure 3.** Flowchart of the optimization procedure.

## 3. Transceiver and Receiver Coil Geometries and Performance Analysis

### 3.1. Transceiver and Receiver Coil Geometries

The geometries of $Rx_1$, $Rx_2$, and $Tx$ were found by repeatedly executing the optimization technique depicted in Figure 4. The diameter of the $Rx_1$ coil, $d_{O1}$, varied with regard to $n_1$, $w_1$, and $p_1$. Similarly, the diameter of the $Rx_2$ coil, $d_{O2}$, varied with regard to $n_2$, $w_2$, and $p_2$, which determined the $d_{O1}$ and $d_{O2}$ as 34 mm. In order to identify optimal turns, $n_1$ and $n_2$ were initially set to 2 and then they were varied between 2 and 6 with respect to $w_1$, $p_1$, and $w_2$, $p_2$. With $n_1 = 3$, the maximum $Q_1$ and self-inductance were 117.55 and 222.08 nH, respectively. Similarly, $Q_2$ and $n_2 = 3$ accomplished the same result. The values $Rx_1$ and $Rx_2$ were simultaneously tuned to maximize $M_{12}$. The value $D_3$ was always set

to 50 mm since the average distance between the heart and the exterior of the chest is 50 mm for practically every human [17–21]. The value $M_{12}$ was greatest at $D_{12} = 0$ mm and gradually dropped as $D_{12}$ increased. The values $Rx_1$ and $Rx_2$ weighed 0.1 g when $n_1 = 3$ and $w_1 = 1$ mm. Due to being lightweight, $Rx_1$ and $Rx_2$ could be effectively administered to the heart. Even if $Tx$ was located above the chest, it is essential to build a small coil. Initial plans for $Tx$ included two turns, but the resulting $M_{13}$ and $M_{23}$ were low. Consequently, $n_3$ was set to 1. The value $Q_3$ and the self-inductance of $Tx$ were 123.2 and 225.1 nH, respectively. Table 1 shows the optimized parameters for $Rx_1$, $Rx_2$, and $Tx$ at 13.56 MHz.

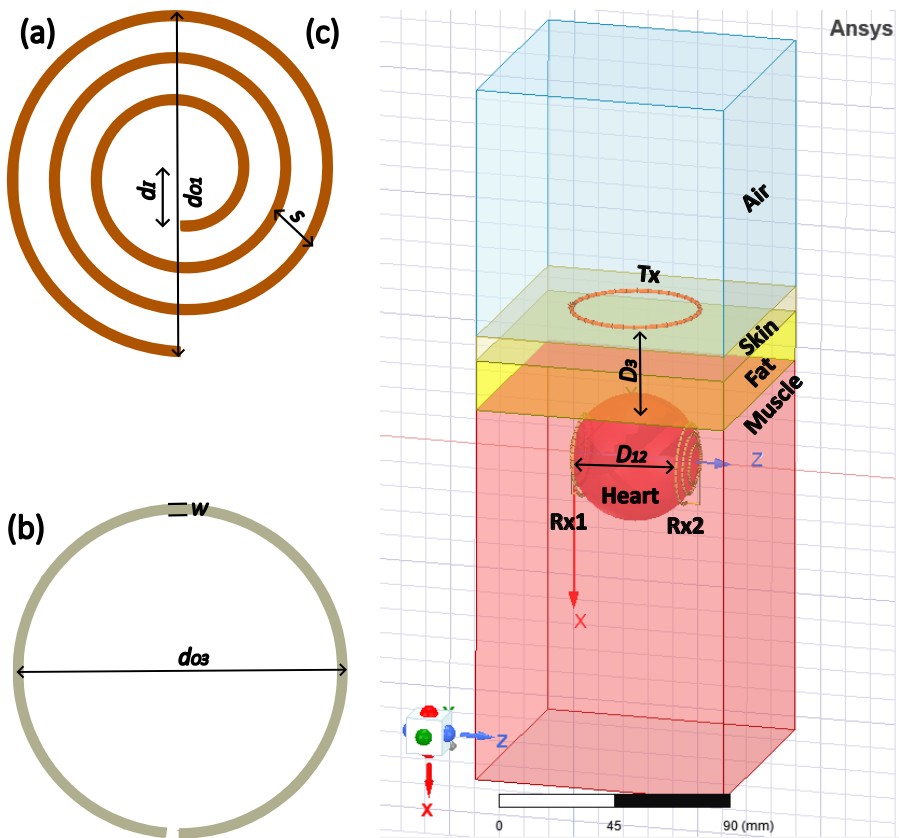

**Figure 4.** (**a**) Receiver $Rx_1$ and $Rx_2$; (**b**) transceiver $Tx$; (**c**) the transceiver coil and receiver coils in the heterogeneous phantom in the simulation study.

**Table 1.** Transceiver and receiver coil specifications at 13.56 MHz.

| Parameter | Symbol | $Rx_1$ | $Rx_2$ | $Tx$ |
|---|---|---|---|---|
| Self inductance (nH) | $L$ | 222.0 | 222.0 | 123.2 |
| Self resistance | $\Omega$ | 0.16 | 0.16 | 0.047 |
| Q-Factor | $Q$ | 117.5 | 117.5 | 225.1 |
| Outer diameter (mm) | $d_O$ | 34 | 34 | 61.5 |
| Inner diameter (mm) | $d_I$ | 10 | 10 | 58.5 |
| Number of turns | $n$ | 3 | 3 | 1 |
| Width (mm) | $w$ | 1 | 1 | - |
| Spacing (mm) | $s$ | 3 | 3 | - |
| Pitch (mm) | $p$ | 1 | 1 | - |
| Weight (g) | - | 0.1 | 0.1 | - |
| Distance (mm) | | $D_{12} = 10$ to $60$ | $D_3 = 50$ | |
| Wire type | | Copper | | |
| Frequency (MHz) | | 13.56 | | |

### 3.2. Performance Analysis in the Heterogeneous Phantom

In order to validate the performance of the inductive coupling method in a real-time environment, we defined a heterogeneous phantom composed of layers of muscle, (dry) skin, fat, and heart. Taking into consideration the actual size of the heart [22–25], the heart layer's thickness was estimated to be 50 mm. The heterogeneous phantom consisted of a thickness of 10 mm for each muscle, skin, and fat layer; however, the heart layer's thickness was estimated to be 50 mm. The geometries of $Rx_1$, $Rx_2$, and $Tx$ in the heterogeneous phantom were employed in the simulation analysis, where $Tx$ resided in the air and $Rx_1$ and $Rx_2$ were located within the heart layer, as depicted in the Figure 4c. As indicated in Table 2, these layers were examined subjected to the loss tangent ($tan\delta$), conductivity ($\sigma$), and relative permittivity ($\epsilon_r$) at 13.56 MHz of the bodily tissue. With $C_1$ = 680 pF, $C_2$ = 680 pF, and $C_3$ = 1000 pF, the resonant frequency in the coils was attained. The values $R_S$ and $R_L$ were set to 100 Ω across $Tx$, $Rx_1$, and $Rx_2$.

**Table 2.** The biological tissues' dielectric characteristics at 13.56 MHz [26].

| Tissue | Relative Permittivity ($\epsilon_r$) | Loss Tangent ($tan\delta$) | Conductivity ($\sigma$) |
|---|---|---|---|
| Air | 1.0 | 0.0 | 0.0 |
| Skin | 285 | 1.10 | 0.23 |
| Fat | 11.8 | 3.40 | 0.03 |
| Muscle | 138 | 6.01 | 0.62 |
| Heart | 239 | 2.91 | 0.52 |

The variations in blood dielectric properties affected the conductivity, so the $S_{33}$ was simulated with different conductivity ranging from 0.6 (S/m) to 1 (S/m). Based on the reported results in Figure 5, at the resonant frequency, a considerable shift in the reflection coefficient was noticed due to the change in conductivity.

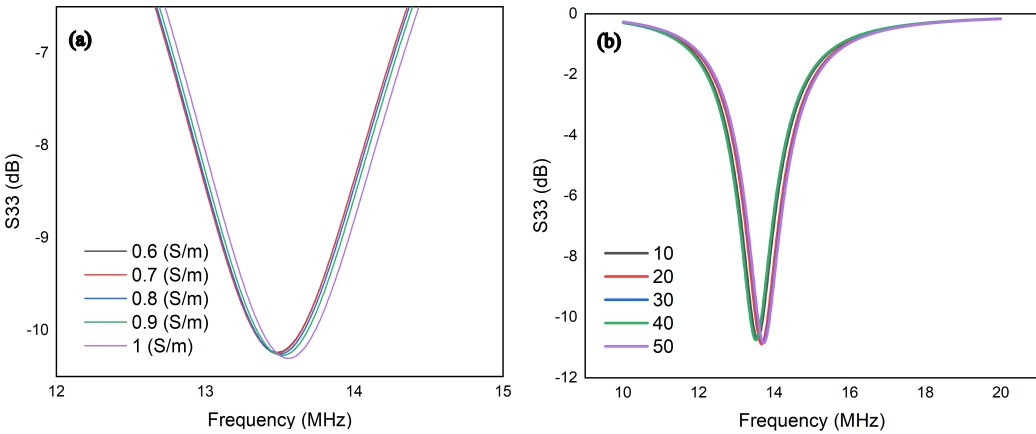

**Figure 5.** (**a**) S33 at different conductivities in the tissue medium under simulation; (**b**) S33 at varied distances in the tissue medium under simulation.

Generally, a human's body is permeable to frequencies under 20 MHz, hence, the allowed exposure limit for frequencies is less than 20 MHz [26,27]. To evaluate the safe operation of the proposed method's SAR, the FCC-mandated basic fundamental exposure limit for whole-body exposure was determined over a tissue mass of 1 g. Conductivity ($\sigma$) and density ($\rho$) of the tissue are directly and inversely proportional to SAR, respectively, and that was estimated with the formula $\sigma|E|^2/\rho$, where $E$ is the electric field [28,29]. The United State's FCC now enforces a SAR limit of 1.6 W/kg over 1 g of tissue for the general population, and South Korea follows the same standard [30]. Japan and the European Union Council propose a level of 2.0 W/kg, over 10 g of actual tissue, on average. Figure 6 demonstrates that the highest average SAR simulated for a heterogeneous phantom was

0.85 W/kg. Therefore, the SAR clearly demonstrated that the value was far less than the permissible W/kg.

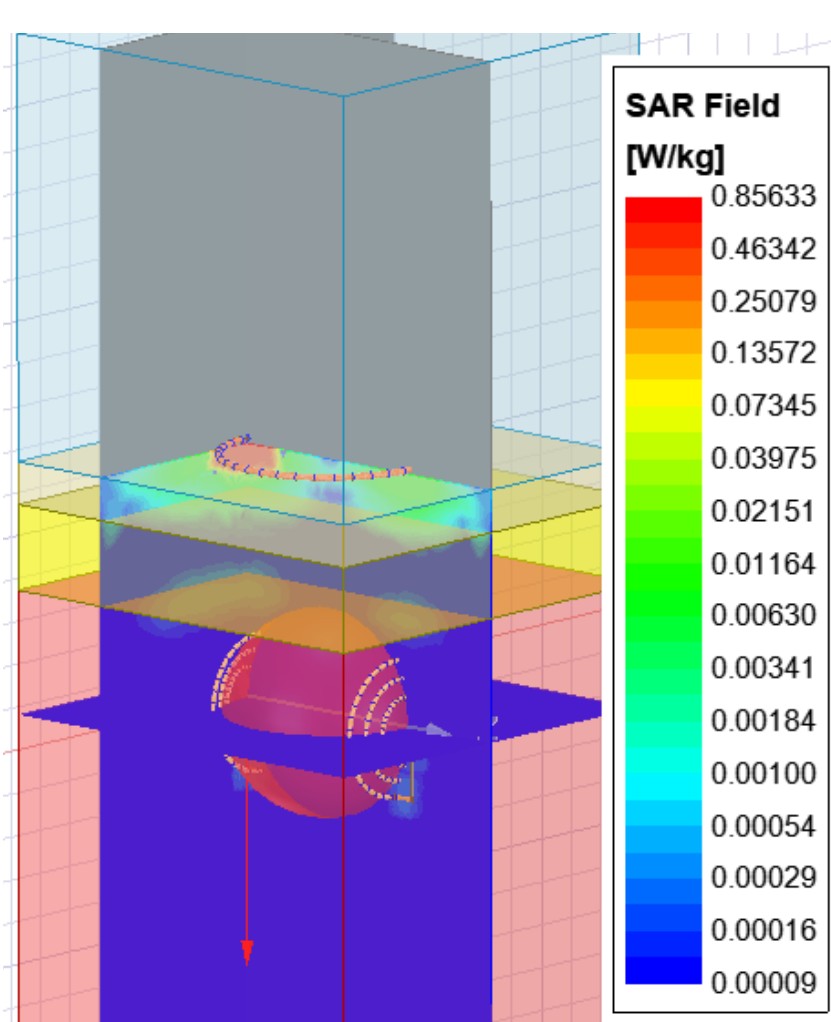

**Figure 6.** Average specific absorption rate (SAR) at the maximum simulated average.

The SAR value varied with $D_{12}$ and it was observed that the SAR for different $D_{12}$, 10 mm, 20 mm, 30 mm, 40 mm, and 50 mm, was also far less than the permissible W/kg.

## 4. Experimental Measurements

In accordance with the optimum specifications derived from the simulation environment, the fabrication procedure was executed. The coils, $Rx_1$ and $Rx_2$, with finalized dimensions from the simulation environment, were printed out and stuck on a plastic bowl in order to trace the copper coil exactly on the surface precisely, as shown in the Figure 7a. The receiver coils were then tuned with capacitance in order to match the resonant frequency. The $Tx$ coil was soldered onto a dotted PCB with a similar procedure (Figure 7b). This $Tx$ coil was then tuned to the resonant frequency. Figure 7c represents the $Rx_1$ and $Rx_2$ placed at a distance of 20 mm in the air medium to compare the changes in $S_{11}$, $S_{22}$, $S_{33}$, and $S_{21}$ between the air and tissue mediums. The entire experimental setup is displayed in Figure 7d in which the $Rx_1$ and $Rx_2$ were dipped into the minced pork, which has similar properties to that of a human heart. The minced pork was refrigerated and placed at room temperature for a few hours before conducting the experiment. The transceiver $Tx$ was placed at exactly 50 mm perpendicular to the receiver coils $Rx_1$ and $Rx_2$. The results were measured using the specified apparatus (Rohde Schwartz, ZVH4, Seoul, Republic of Korea).

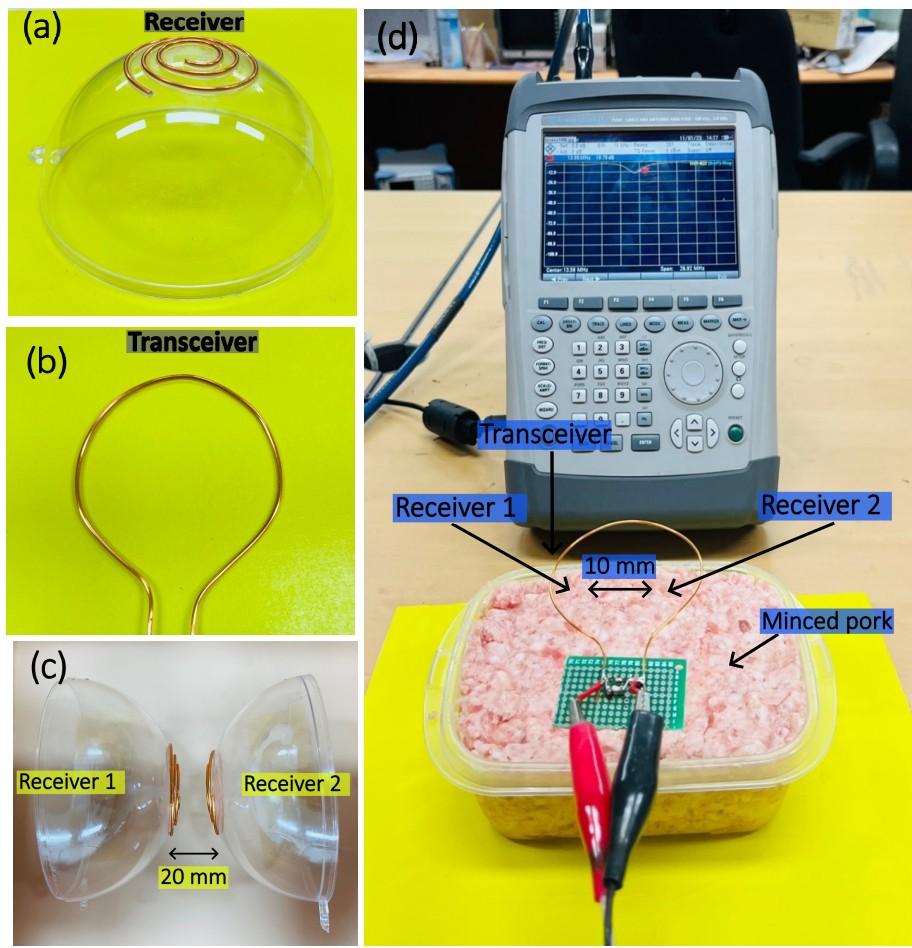

**Figure 7.** (**a**) The spherical receivers, $Rx_1$ and $Rx_2$, with diameters of 34 mm; (**b**) Planar $Tx$ with a diameter of 61 mm; (**c**) $Rx_1$ and $Rx_2$ placed in the air medium with $D_{12}$ at 20 mm; (**d**) the experimental setup for measuring $S_{33}$ under different conditions.

To determine the $S_{11}$, $S_{22}$, and $S_{33}$, the $Rx_1$, $Rx_2$, and $Tx$ were driven by port 1, which were tuned by the capacitances $C_1$, $C_2$, and $C_3$, respectively. The obtained $S_{11}$, $S_{22}$, and $S_{33}$ values were $-20.8$ dB, $-25.4$ db, and $-16.6$ dB, respectively. Variations in $S_{33}$ in the tissue medium imply that the transceiver coil placed outside the minced pork could be used to monitor changes in the distance between the implanted receiver coils, hence predicting the shift in the distance, as shown in Figure 8. Similarly, for the real-time application, the same procedures will be carried out. The coils, on the other hand, will be made of flexible PCB instead of the copper coils that were used in the experiments. This is because flexible PCB is very light and gives the patient a greater degree of comfort. In addition, when the system is used in the real world, a nanoVNA will be attached to the transceiver/external coil so that the entire system is small and it is easy to use and gauge changes in the receiver coils.

*Comparison between Simulated and Measured Results*

$S_{11}$, $S_{22}$, and $S_{12}$ were also evaluated in both the measured and simulated results in order to compare the effectiveness of the implanted coils in all aspects. Measuring the changes in $S_{33}$ is the most important factor for monitoring the size of the ventricular chamber as it can be operated without any physical contact with the human body since it is placed above the chest connected to a nanoVNA. Comparing the results revealed that both the simulated and measured data exhibited a similar trend. Interestingly, In Figure 9c, the $S_{33}$ in the simulated result followed a similar pattern to $S_{33}$ in the measured result with a difference of about $-0.4$ dB. As previously indicated, although $S_{11}$ and $S_{22}$ (Figure 9a,b) displayed a substantial divergence between the simulated and measured

values, both $S_{11}$ and $S_{22}$ grew linearly with increasing distance and exhibited a dramatic drift in the reflection coefficients at 20 mm and 30 mm. Therefore it is clear from Figure 9c that $S33$ demonstrated shifts in the distance between the implanted coils, and this could be easily supervised by placing the transceiver coil just above chest in order to examine the expansion and contraction of the heart, as the reflection coefficient varies depending on the distance between the left and right ventricular chambers.

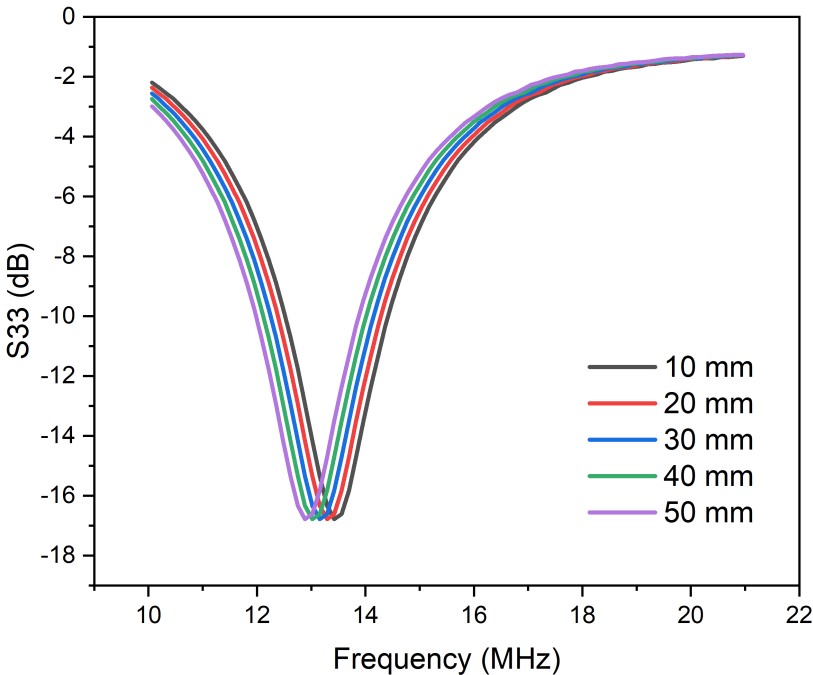

**Figure 8.** $S_{33}$ measured at different distances in the tissue medium.

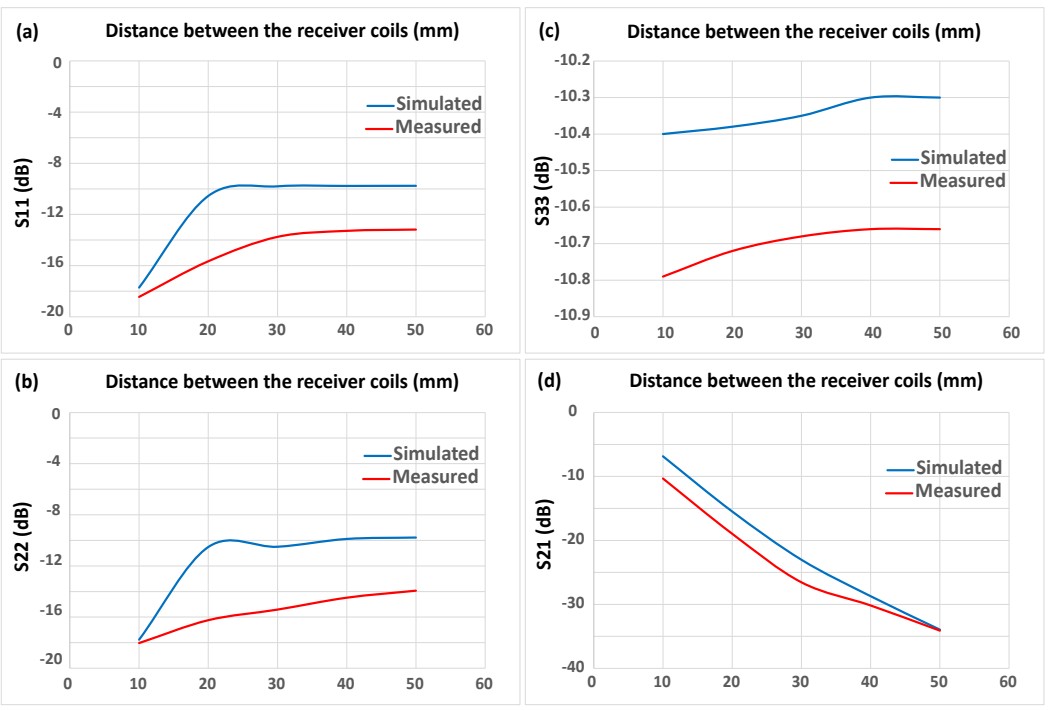

**Figure 9.** Comparison between the simulated and the measured results in the tissue medium: (**a**) $S_{11}$, (**b**) $S_{22}$, (**c**) $S_{33}$, (**d**) $S_{21}$.

## 5. Conclusions

Heart failure can be predicted based on a number of different factors, such as the expansion and contraction of the heart as well as the quantity of fluid that is present in the heart. It was found that a change in the reflection coefficient at the resonant frequency of the transceiver coil represented a change in distance between the chambers of the heart. Although the fluids present in the heart can be determined using this method, the expansion and contraction of the heart are considered to be the most critical aspect of this research. In the experimental results, the $S_{33}$ with a 50 mm distance between the implanted coils demonstrated a reflection coefficient of $-10.66$ dB at resonant frequency whilst the $S_{33}$ with a 10 mm spacing between the implanted coils exhibited a reflection coefficient of $-10.79$ dB at the resonant frequency, and this shift in the value clearly indicated that the variation in the implanted coils' distance apart, with pork skin placed in between, influenced the reflection coefficient of the transceiver coil at the resonant frequency.

In the future, a magneto-resistive sensor (TMR) can be employed to detect the position of the implanted coils. This TMR sensor can be placed between the transceiver coil and the implanted coils and can also be used to evaluate coil misalignment due to the simplicity of the computation procedure by measuring the mutual inductance [13,31]. This variation in the mutual inductance between the transceiver and receiver coils will provide the distance between the coils more efficiently. The mutual inductance increases with an increased distance between receiver coils since the coils will be aligned perpendicularly to the transceiver when the distance is varied. The mutual inductance suddenly drops down when the receiver coils surpass the transceiver coil's boundary. Hence, this idea will be implemented in the near future by involving ESP32WROOM-32E MCU and the MIT app inventor, which provides data to a handheld device via Bluetooth 2.45 GHz.

**Author Contributions:** Conceptualization, J.R.C.; methodology, R.K.V. and K.M.; simulation, J.L.; writing—original draft, R.K.V. and K.M.; review, J.R.C., R.K.V., V.G. and K.Y.; supervision, J.R.C.; project administration, J.R.C.; funding acquisition, J.R.C. All authors have read and agreed to the published version of the manuscript.

**Funding:** This work was supported by the National Research Foundation of Korea (NRF) grant funded by the Korea government (MSIT) (No. 2020R1I1A3065961) and BK21 Four project funded by the Ministry of Education, Korea (No. 4199990113966).

**Institutional Review Board Statement:** Not applicable.

**Informed Consent Statement:** Not applicable.

**Conflicts of Interest:** The authors declare no conflict of interest.

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
