# Peer review of "Design and Analysis of an Inductive Coupling System for the Early Detection of Heart Failure"

_applsci, doi:10.3390/app13074381_

Round 1

Reviewer 1 Report

Authors have submitted a novel on "Design and Analysis of an Inductive Coupling System for Early Detection of Heart Failure". The following points to be noted

1. The current work is not compared with existing work. Justify?

2. How do you calculated artifacts in S33 parameters?

3. Is it S33  value is enough to calculate expansion?

4. Have you consider the conductivity of skin , muscle  in the design of antenna? if yes what is significance in the result. 

Author Response

Please find the author response in the attached file.

Reviewer 2 Report

The acticle needs to be majorly revised.

Author Response

(The authors gave the same response as above.)

Reviewer 3 Report

This paper proposes a method to sense the current state of the heart by inserting two circuit-less 30 mm spherical receiver coils implanted in both the left and right sides of the heart. They are flexible spherical receiver coils on either side of the ventricular chambers and positioning a transceiver coil directly above the chest, which is done to observe the changes in the resonant frequency and the reflection coefficient of the external coil. The research is inspired to improve the detection of heart failure employing hemodynamic parameter evaluation.

There are some suggestions that authors must take into consideration to improve the quality of the paper:

-       The abstract should be rewritten, highlighting the main contributions and emphasizing the motivation of this study.

-       The “Introduction” must be rewritten to motivate the readers in its lecture. It is necessary to present the problem, the benefits of this technology, and how it can support the research problem.

-       It is also essential to create a “new section 2” that contains the literature review regarding the thematic research and related works, analyzing those research works with your proposed methodology.

-       Conclusions must be rewritten, emphasizing the results and contribution. Moreover, future works should be pointed out in this section.

Author Response

(The authors gave the same response as above.)

Round 2

Reviewer 2 Report

Moderate English changes required.